# OpenReview forum: "EFFICIENT JAILBREAK ATTACK SEQUENCES ON LARGE LANGUAGE MODELS VIA MULTI-ARMED BANDIT-BASED CONTEXT SWITCHING"
_ICLR.cc/2025/Conference — ICLR 2025 Poster_

### Official Review · Reviewer_pzyz · 2024-10-17

**Soundness:** 3
**Presentation:** 3
**Contribution:** 3
**Rating:** 6
**Confidence:** 4

**Summary:**

This paper proposes a novel method for jailbreaking large language models (LLMs) through "Sequence of Contexts" (SoC) attacks and utilizes a multi-armed bandit (MAB) framework to automate the optimization of the attack process. The paper also provides an in-depth theoretical analysis of the sequence length required for a successful jailbreak and the convergence of total rewards.

**Strengths:**

The study not only experimentally demonstrates the efficiency of the proposed method, achieving an attack success rate of over 95%, but also provides a solid theoretical foundation for LLM jailbreak attacks.

**Weaknesses:**

1. It is a natural question: why do you not compare your work with other methods，such as GCG[1] , AutoDAN[2], PAIR[3], RENELLM[4] and so on?
2. Since your work requires selecting a sequence of context-switching queries, I am curious about it time complexity.
3. In my opinion, testing only on Mistral and Llama is not sufficient to demonstrate the advantages of your work. Moreover, you have only chosen small LLMs (up to 8B), which is not convincing. As far as I know, Llama has a 13B version. What's more, CHATGPT is necessary to be choose.
In conclusion, without comparisons, it is difficult for me to fully assess the contributions of this paper, especially considering that there are many papers on LLM jailbreaks. If you address my concerns, I will consider giving a higher score.

Ref:
[1] Universal and transferable adversarial attacks on aligned language models.
[2] Autodan: Generating stealthy jailbreak prompts on aligned large language models.
[3] Jailbreaking black box large language models in twenty queries.
[4] A Wolf in Sheep’s Clothing: Generalized Nested Jailbreak Prompts can Fool Large Language Models Easily

**Questions:**

See weekness.

---

> ### Author Response · Authors · 2024-11-20
> **Rebuttal Part (1/1)**
>
> We sincerely thank you for your valuable comments, observations, and the time you have dedicated to evaluating our paper. We have carefully addressed your questions and the weaknesses you highlighted in a comprehensive manner. Our response is structured by presenting each weakness or question posed, followed by our detailed reply.
>
> **Weakness 1:**  It is a natural question: why do you not compare your work with other methods such as GCG[1] , AutoDAN[2], PAIR[3], RENELLM[4] and so on?
>
> **Response-to-Weakness-1:**
>
> In response to reviewer GXrd we compare our proposed Jailbreak attack method to the current SOTA jailbreak methods. We request the reviewer to refer to **Table RT2** in **Response-to-Weakness-3** to the reviewer **GXrd**.
>
> **Weakness 2:** Since your work requires selecting a sequence of context-switching queries, I am curious about it time complexity.
>
> **Response-to-Weakness-2:**
>
> - In the proposed jailbreak attack method, at each step the MAB decision making framework chooses a context-switching-action  using an $\epsilon$-greedy strategy by balancing exploration of new context-switching-actions and exploitation of existing context-switching-actions. (See Line 13 in Algorithm-1 on Page 4 of the main paper). This can be done in constant time $O(1)$ by using efficient data structures such as priority-queues at each step. Thus, for the entire multi-turn attack of $T$ rounds, the time-complexity is $O(T)$.
>
> - We request the reviewer to refer to the response **Response-to-Weakness-3** for the reviewer **GXrd**, where we compare the computational cost of our methods with various SOTA jailbreak attack strategies.
>
> **Weakness-3:** In my opinion, testing only on Mistral and Llama is not sufficient to demonstrate the advantages of your work. Moreover, you have only chosen small LLMs (up to 8B), which is not convincing. As far as I know, Llama has a 13B version. What's more, CHATGPT is necessary to be choose. In conclusion, without comparisons, it is difficult for me to fully assess the contributions of this paper, especially considering that there are many papers on LLM jailbreaks. If you address my concerns, I will consider giving a higher score.
>
> **Response-to-Weakness-3:**
>
> We agree that evaluation of our method on proprietary models such chatGPT and larger open-source models such as Llama-13B will further help us assert our claim of high ASR of our proposed method. In this regard, we have evaluated our method on Llama-2-13B-chat and Vicuna-13B- v1.5.
>
> | LLM Model          | ASR (%) |
> |---------------------|---------|
> | Llama-2-13B-chat    |   0.905      |
> | Vicuna-13B- v1.5  	| 0.902  	|
>
> **Table RT 5:** Performance of our attack on Llama-2-13B and Vicuna-13B- v1.5.
>
> Our method is able to successfully jailbreak chatGPT from our preliminary experiments. However, the harmful responses generated by chatGPT is triggering the output content filter in the chatGPT-API and this blocks us from asking further questions. We are in the process of generating results for chatGPT and will add the performance on chatGPT shortly.
>
> In summary the our contributions to address the reviewer's concerns include :
>
> - Comparison of the proposed jailbreak method with other state-of-the-art (SOTA) methods in terms of Attack Success Rate (ASR) and Cost Per Prompt (CPP).
> - Clarifications regarding the time complexity of the proposed method.
> - Experiments on larger LLMs using our method to validate our claim of achieving a high Attack Success Rate (ASR).
> - Enhancements to improve the presentation and readability of the paper.
>
> We hope the technical clarifications, additional experiments, and justifications we provided have addressed your concerns. If you find the responses satisfactory, we kindly request you to consider raising the score above the acceptance threshold. Please let us know if there are any remaining issues or additional details you would like us to address, which we would do it right away.

---

> > ### Comment · Reviewer_pzyz · 2024-11-25
> >
> > Since I have run the baseline jailbreak attacks, I am doubtful about your result, which is much lower than I know.
> > Besides, there is a website about the jailbreak benchmark[1].
> >
> > [1]. https://jailbreakbench.github.io/

---

> ### Author Response · Authors · 2024-11-26
> **Clarification For Baseline Evaluation**
>
> We would like to respectfully clarify to the reviewer that our baselines were evaluated on a different
> dataset compared to the one referenced on the website [1]. This difference can significantly impact the
> ASR results for various methods. Our dataset has 2700 harmful questions (See section 5.1 in the main draft) compared to the dataset mentioned in the website which has only 100 questions. Additionally, the language models used on the website are older versions of the LLaMA models, specifically the LLaMA-2 model.
>
> It is important to note that LLaMA models are continuously updated with improved safety fine-tuning,
> making newer versions more resilient to previously proposed jailbreak attacks. This results in a significant
> decrease in ASR for the latest model iterations. In our work we mainly used Llama-3 models which have better safety fine tuning compared to the older versions of the model.
>
> We kindly request the reviewer to refer to Table 2 in [2] and Table 1 (No defence, LLaMA-2 scores)
> in [3], which present results for the same methods we have compared against. These tables demonstrate that the ASRs for  the referenced methods are either lower than or comparable to those in our work.
>
> [2] https://arxiv.org/pdf/2311.08268#page=5.65  (ReNeLLM)
>
> [3] https://aclanthology.org/2024.acl-long.303.pdf#page=6.62 (Safe Decoding)

---

> > ### Comment · Reviewer_pzyz · 2024-11-26
> >
> > Thank you for your reply. I will raise the score. In addition, I still think that experiments based on ChatGPT or similar closed-source large models are necessary. I recommend that the author complete the experiments before submitting the paper, so as to better contribute to the community.

---

> > > ### Author Response · Authors · 2024-11-26
> > > **Response to Reviewer Suggestion**
> > >
> > > Thank you for your thoughtful response and for raising the score. We sincerely appreciate your suggestion regarding experiments with ChatGPT or similar closed-source large models.
> > >
> > > As part of our efforts, we have conducted experiments using Cohere, a closed-source LLM. Cohere provides an option to turn off the content filter, which allowed us to perform our experiments without interference.
> > >
> > > | Model   | ASR  |
> > > |---------|------|
> > > | Cohere  | 0.91 |
> > > **Table RT6:** Performance of our attack on Cohere's closed source LLM.
> > >
> > > For ChatGPT, we are still working to complete the experiments, but progress has been challenging due to repeated blocks by the content filter. We are actively exploring ways to overcome these limitations to provide a more comprehensive evaluation.
> > >
> > > Given the significant updates we have made to address your concerns, we kindly request you to consider raising the score above the acceptance threshold if you find the revisions satisfactory. We are committed to further improvements as needed and deeply value your feedback and support.

---

> > > > ### Comment · Reviewer_pzyz · 2024-11-27
> > > >
> > > > Thanks to the author's supplementary experiments, I have a more comprehensive understanding of the article. This is a great piece of work and I will raise the score.

---

### Official Review · Reviewer_rSU9 · 2024-10-28

**Soundness:** 1
**Presentation:** 1
**Contribution:** 2
**Rating:** 5
**Confidence:** 3

**Summary:**

This paper discovers the LLMs intend to reject direct malicious queries (DMQs), but answer when followed by content switching queries (CSQs). For automatically generating DMQs and CSQs, the authors propose a framework based multi-armed bandit (MAB) to jailbreak LLM automatically. They introduce a dataset of CSQs based on MAB. And they give a mathematical derivation for the proposed method to prove key bounds.

**Strengths:**

### 1. The discovery about switching the context leads jailbroken
This paper presents a context switching attack, which is a novel method compared to existing works. And combined with multi-armed bandit, SoC can automatically jailbreak LLMs.


### 2. Theoretical Results
Section 4 establishes a upper bound on the length of SoC attack sequence. And I believe a method which has a mathematical proof is more solid than existing jailbreak works.

**Weaknesses:**

### 1. Poor readability
In the abstract and in the overview (Figure 1) , the authors mention "multi-armed bandit (MAB)", but they do not explain what is MAB in the introduction.I suggest the authors revise this, it confuses me until I have read related works.

In addition to MAB, the introduction has unreasonable content. With four paragraphs, more than half introduction is irrelevant to the contribution of this paper. And in the third paragraph, there are too many concepts are proposed but lack of details. I suggest that the authors adjust their introduction, current version has poor readability.

I think it is not appropriate to introduce how this paper uses MAB in related work. And why is T both rounds and sequence length (*total reward over T rounds*: line133; *attack sequence length T*:line137)? What does T represent in the subsequent content?

There are a lot of abrupt concepts introduced without much explanation in context. In line 215, I can not find any information or reference about *policy $\pi$* and *action-value Q* (including cost C in line 240). I can only speculate that this has something to do with reinforcement learning. And in Algorithm 1, what does  $E_{explore}$ stand for and what does  $E_{exploit}$ stand for? I can not find any explanation.

### 2. Lack of comparison
This paper only demonstrate SoC can jailbreak LLMs, but has no comparison with prior works. I believe that there are many excellent baseline jailbreak methods. The SoC attack has similar format with the In-Context Attack[8] which also jailbreak LLMs based on context. Besides, the authors optimize their method with reinforcement learning, but the binary reward is very similar to PAIR, which use LLM optimize jailbreak prompt[9].


### 3. Few experiments
In the entire paper, only four figures in Figure 3 prove that SoC is effective, and there is a lack of comprehensive experiments from multiple angles. For example, with different hyperparameters such as J and K, I am not sure whether this method can be generalized or only performs well under certain specific parameters.

---

[8]  Jailbreak and guard aligned language models with only few in-context demonstrations

[9] Jailbreaking Black Box Large Language Models in Twenty Queries

**Questions:**

### 1. Related Works
I believe that in the `White-Box Attack`, the authors should not cite a lot of attack but is irrelevant to jailbreak. Besides GCG, there are many white-box attack, such as fine-tuning attack[1, 2], improved GCG[3], interpretability-based[4, 5]. Nevertheless, I am only making a suggestion. Whether the author makes modifications or not will not change my rating.

### 2. Why does the authors think modern LLMs avoid responding to malicious questions by classifier?
In Section 3.1 line 155-line157, the authors mention "In most instances, such queries fail to produce harmful responses and can be guarded using straightforward strategies, such as employing a classifier to flag harmful words and phrases". Current LLMs do not use those filter, but are fine-tuned to align with human values[6, 7].

### 3. There is no update for $\pi$ or $\pi^{*}$
Algorithm 1 aims to optimize a policy $\pi$, however, where is update for $\pi$ or $\pi^{\star}$. Why can Algorithm 1 obtain a optimized policy $\pi^{\star}$? This confuses me as to how Soc Attack works.

### typos
1. Incorrect use of quotation marks: line 153-155,
2. Do B and C refer to the appendix? line 201 & 202 & line 318 & line 321 & line 365
3. Do you mean harmful or harmless? line 213-214: *which assigns a binary reward indicating whether the response is harmful or unsafe.*
4. *see-D.0.1*, *see-D.0.2*, *see-D.0.3* and *see-D.0.4* mean Appendix D.0.1, Appendix D.0.2, Appendix D.0.3 and Appendix D.0.4?

---


[1] FINE-TUNING ALIGNED LANGUAGE MODELS COMPROMISES SAFETY EVEN WHEN USERS DO NOT INTEND TO!

[2] SHADOW ALIGNMENT: THE EASE OF SUBVERTING SAFELY-ALIGNED LANGUAGE MODELS

[3] AmpleGCG: Learning a Universal and Transferable Generative Model of Adversarial Suffixes for Jailbreaking Both Open and Closed LLMs

[4] Uncovering Safety Risks in Open-source LLMs through Concept Activation Vector

[5] How Alignment and Jailbreak Work: Explain LLM Safety through Intermediate Hidden States

[6] Training language models to follow instructions with human feedback

[7] Training a Helpful and Harmless Assistant with Reinforcement Learning from Human Feedback

---

> ### Author Response · Authors · 2024-11-20
> **Rebuttal Part (1/5)**
>
> We sincerely thank you for your valuable comments, observations, and the time you have dedicated to evaluating our paper. We have carefully addressed your questions and the weaknesses you highlighted in a comprehensive manner. Our response is structured by presenting each weakness or question posed, followed by our detailed reply.
>
>
> **Weakness-1:** In the abstract and in the overview (Figure 1) , the authors mention "multi-armed bandit (MAB)", but they do not explain what is MAB in the introduction. I suggest the authors revise this, it confuses me until I have read related works.
>
> **Response-to-Weakness-1:**
>
> We sincerely thank the reviewer for their insightful comment regarding the introduction of the MAB concept in our manuscript. We appreciate your feedback, and we will revise the draft to provide a clear and concise introduction to MAB in the suggested sections. This addition should make the concept and its relevance to our work more accessible from the outset.
>
>
> **Weakness-2:** In addition to MAB, the introduction has unreasonable content. With four paragraphs, more than half introduction is irrelevant to the contribution of this paper. And in the third paragraph, there are too many concepts are proposed but lack of details. I suggest that the authors adjust their introduction, current version has poor readability.
>
> **Response-to-Weakness-2:**
>
> Thank you for your valuable feedback on our paper's introduction. We appreciate your insights regarding the relevance and readability of the introductory section. We understand the importance of maintaining a focused and clear introduction that directly supports our contributions. Based on your suggestion, we will adjust the introduction to ensure that all content aligns more closely with the paper's main contributions and reduces any extraneous information. Additionally, we will refine the third paragraph to provide clearer explanations and avoid introducing too many concepts without adequate detail.
>
>
> **Weakness-3:** I think it is not appropriate to introduce how this paper uses MAB in related work. And why is T both rounds and sequence length (total reward over T rounds: line133; attack sequence length T:line137)? What does T represent in the subsequent content?
>
> **Response-to-Weakness-3:**
>
> Thank you for pointing out the notation ambiguity regarding $T$ in our description of the MAB framework and its use in the training algorithm. We acknowledge that our use of $T$ as both the total number of episodes and as a reference to the context length during each episode may cause confusion.
>
> To clarify, $T$ in our setting represents the number of episodes over which we want to maximize the total reward obtained during the course of the SoC attack, each corresponding to one interaction in the training phase. Within each episode, the "length" of the context window dynamically increases, as shown in line 16 of the training algorithm. This context window accumulates as more context-switching queries (CSQs) are appended to it.

---

> ### Author Response · Authors · 2024-11-20
> **Rebuttal Part (2/5)**
>
> **Weakness-4:**  There are a lot of abrupt concepts introduced without much explanation in context. In line 215, I can not find any information or reference about policy $\pi$ and action-value Q (including cost C in line 240). I can only speculate that this has something to do with reinforcement learning. And in Algorithm 1, what does $E_{explore}$ stand for and what does $E_{exploit}$ stand for? I can not find any explanation.
>
> **Response-to-Weakness-4:**
>
> Thank you for your valuable feedback. We appreciate the opportunity to clarify the concepts introduced in our paper and address the points you've raised.
>
> 1. **Explanation of $E_{explore}$ and $E_{exploit}$:**
>
>    - In the training algorithm, $E_{explore}$ and $E_{exploit}$ represent the counts of exploration and exploitation actions taken, respectively. These counters track how often we engage in exploration (choosing a random action) versus exploitation (selecting the best-known action based on the current Q-values). They are used to compute the cost function which updates the action values across episodes, which is a key aspect of the epsilon-greedy strategy (See Line 18 in Algorithm 1 on Page 4 in the main paper).
>
>
>
> 2. **Relation between Policy $\pi$ and Q-value $Q(a)$:**
>
>    - The policy $\pi$ in this context is essentially a decision rule that dictates which action (CSQ category $a$) to select at any given step. The Q-value $Q(a)$ represents the expected reward for choosing action $a$ (a particular CSQ category) given the current knowledge.
>
>    - The training algorithm iteratively updates the Q-value for each action using the observed rewards, enabling the model to learn the optimal action-value function $Q(a)$ for each DMQ category.
>
>    - The optimal policy $\pi^*$ is a function of the Q-values and is defined as $\pi^{*} = \arg\max_{a}(Q(a))$. This policy selects the action that maximizes the expected reward according to the learned Q-values at each step of the SoC attack, ensuring that the algorithm converges to a strategy that maximizes harmful or unsafe responses over time.
>
>
>
> 3. **Clarification of Training Algorithm’s Purpose:**
>
>    - The training algorithm is designed to learn these Q-values $Q(a)$ by balancing exploration and exploitation. Over multiple episodes, the Q-values for each CSQ category get refined based on the observed rewards, allowing the model to converge toward an optimal policy for SoC  -based jailbreak attacks.
>
>
> We understand that the introduction of certain concepts may have seemed abrupt without additional context, and we apologize for any confusion caused. We will make sure to revise the draft to clearly define $E_{explore}$, $E_{exploit}$, the relationship between policy $\pi$ and Q-value $Q(a)$, and the role of the training algorithm in learning an optimal policy.
>
> Your insights have been incredibly helpful in improving the clarity of our paper. Thank you once again for helping us enhance the readability and precision of our work.

---

> ### Author Response · Authors · 2024-11-20
> **Rebuttal Part (3/5)**
>
> **Weakness-5:**  This paper only demonstrate SoC  can jailbreak LLMs, but has no comparison with prior works. I believe that there are many excellent baseline jailbreak methods. The SoC  attack has similar format with the In-Context Attack[8] which also jailbreak LLMs based on context. Besides, the authors optimize their method with reinforcement learning, but the binary reward is very similar to PAIR, which use LLM optimize jailbreak prompt[9].
>
> **Response-to-Weakness-5:**
>
> In response to  reviewer GXrd, we detail and interpret the comparison with existing SOTA jailbreak methods. We request the reviewer to refer to the **Table RT 2** in **Weakness-3** and the corresponding response **Response-to-Weakness-3**. We additionally also detail and compare the existing jailbreak methods based on their computational costs.
>
> Previous multi-turn jailbreak attacks, such as the In-Context Attack, have manipulated the LLM into a jailbroken state by leveraging the context. However, we argue that these attacks fall under the category of templatized approaches and face limitations that we address in our work.
>
> For instance, as noted in Section 3.2 of the In-Context Attack paper, the method relies on sequences of handcrafted prompts, which are often manually created to attempt to jailbreak the LLM. These sequences may include unnecessary or irrelevant steps, leading to longer attack sequences. In contrast, our approach learns a policy within a multi-armed bandit framework, balancing exploration of new context-switching actions and exploitation of existing ones using an $\epsilon$-greedy strategy. This allows us to minimize the attack sequence length by intelligently selecting context-switching actions.
>
> We agree with the reviwer’s comment on the similarities of our binary reward with PAIR. However, we would like to emphasize the following issues with PAIR that we have addressed in our work.
>
> While PAIR employs a binary reward function to indicate whether the LLM has been successfully jailbroken and does not optimize to acheive jailbreak in a minimum number of steps, our approach leverages this function to estimate the expected average reward for each context-switching action. This allows us to learn an optimal policy that minimizes the number of steps required to achieve a jailbreak.
>
> Additionally, PAIR relies on an LLM to refine input prompts based on conversation and reward history, which can increase the number of steps needed for a successful jailbreak. This approach may lead to misaligned prompts without a clear objective, further hindering its efficiency.
>
>
> We request the reviewer to refer to **Table RT2** in our reponse **Response-to-Weakness-3:** to reviewer **GXrd** where, we demonstrate the gain acheived by our method in terms of CPP compared to other SOTA jailbreak methods.
>
> **Weakness-6:**  In the entire paper, only four figures in Figure 3 prove that SoC  is effective, and there is a lack of comprehensive experiments from multiple angles. For example, with different hyperparameters such as J and K, I am not sure whether this method can be generalized or only performs well under certain specific parameters.
>
> **Response-to-Weakness-6:**
>
> In response to reviewer feedback and to further assert our claims, we have carried out additional experiments focusing on
>
> The generalization capabilities of our model, wherein we perform 5 fold cross validation by performing policy training on a subset of the original DMQ categories and test on unseen DMQ categories. We find that in terms of ASR our methods achieve’s performance, comparable to training with all the DMQ categories, with the drop in performance being marginal. We request the reviewer to please refer to **Table RT1** in our response to **Weakness-2)** of the reviewer **GXrd**.
>
> A comparison of the judge function used in our work with the current SOTA judge functions from the Llama-Guard Family. We request that the reviewer refer to **Table RT3** in our response **Response-to-Question-2:** to the reviewer **GXrd**.
>
> In our method the 3 main hyperparameters are the weights asSoC iated with the cost function which guides the MAB policy learning $w1$, $w2$ and the exploration parameter $\epsilon$ which controls the how our our method balances exploration and exploitation of the context-switching-actions. To choose the best values of the aforementioned hyperparameters, we performed a grid search over the hyperparameters by using a binary search approach to arrive at the hyperparameter values that yield the best performance in terms of ASR in a short amount of time.

---

> ### Author Response · Authors · 2024-11-20
> **Rebuttal Part (4/5)**
>
> **Question-1:**    I believe that in the White-Box Attack, the authors should not cite a lot of attack but is irrelevant to jailbreak. Besides GCG, there are many white-box attack, such as fine-tuning attack[1, 2], improved GCG[3], interpretability-based[4, 5]. Nevertheless, I am only making a suggestion. Whether the author makes modifications or not will not change my rating.
>
>
> **Response-to-Question-1:**
>
> Thank you for your feedback and for suggesting additional relevant literature on white-box attacks, including fine-tuning and interpretability-based attacks. We appreciate your suggestion and will incorporate these references into our draft to ensure comprehensive coverage of existing methodologies.
>
> Regarding the references cited in our paper, we believe all cited works contribute to the broader understanding of jailbreak attacks, including relevant aspects of the context and methodology we explored. However, we welcome any specific feedback on particular references that you feel may be irrelevant. This will help us refine our citations to enhance clarity and relevance further.
>
>
> **Question-2:** In Section 3.1 line 155-line157, the authors mention "In most instances, such queries fail to produce harmful responses and can be guarded using straightforward strategies, such as employing a classifier to flag harmful words and phrases". Current LLMs do not use those filter, but are fine-tuned to align with human values[6, 7].
>
> **Response to Question-2:**
>
> To address the reviewer’s query, we clarify that the statement refers to the common practice of wrapping LLM-APIs with guardrails that perform content filtering to prevent responses to prompts containing harmful phrases. This serves as an additional safety layer alongside the safety fine-tuning achieved through RLHF.
>
> **Question-3:** Algorithm 1 aims to optimize a policy $\pi$, however, where is update for $\pi$ or $\pi^{\star}$. Why can Algorithm 1 obtain a optimized policy $\pi^{\star}$? This confuses me as to how SoC  Attack works.
>
> **Response-to-Question-3:**
>
> Thank you for your valuable feedback and for highlighting this point of potential confusion.
>
> 1. **Clarification on Policy Optimization**: In the Multi-Armed Bandit (MAB) setting we are using, the policy $\pi$ itself is not directly updated. Instead, our algorithm uses a standard $\epsilon$-greedy approach, where the Q-values $Q(a)$ (the expected reward for each action $a$, or CSQ category in this case) are iteratively updated based on the rewards obtained during exploration and exploitation. The policy $\pi$ in this context is defined implicitly by the learned Q-values. Once training is complete, the optimal policy $\pi^*$ is obtained by selecting the action $a$ that maximizes $Q(a)$, i.e., $\pi^* = \arg\max_{a} Q(a)$. This means that the optimal policy $\pi^*$ is derived from the Q-values, not directly updated as in traditional reinforcement learning.
>
>
>
> 2. **Acknowledgment of Ambiguity**: We respectfully agree with your observation that the algorithm is presented ambiguously, which could lead to misunderstandings about how the policy $\pi$ is derived. We will revise the draft to clarify that in the MAB framework, the Q-values are updated, and the optimal policy $\pi^*$ is implicitly derived from these Q-values rather than being directly optimized.
>
>
> **Typos:**
>
> 1. Incorrect use of quotation marks: line 153-155,
> 2. Do B and C refer to the appendix? line 201 & 202 & line 318 & line 321 & line 365
> 3. Do you mean harmful or harmless? line 213-214: which assigns a binary reward indicating whether the response is harmful or unsafe.
> 4. see-D.0.1, see-D.0.2, see-D.0.3 and see-D.0.4 mean Appendix D.0.1, Appendix D.0.2, Appendix D.0.3 and Appendix D.0.4?
>
> **Response-to-Typos:**
>
> 1. We thank the reviewer for pointing this out, we will fix this in the draft.
> 2. Yes, B and C refer to the appendix sections.
> 3. To clarify, we aim to state that the judge function $J$ assigns a reward of 0 when the input to it is safe/harmless and assigns a reward of 1 when the input to it is unsafe/harmless. We request the reviewer to look at section B.4 in the appendix for more details.
> 4. Yes they refer to the corresponding sections in the appendix.

---

> ### Author Response · Authors · 2024-11-20
> **Rebuttal Part (5/5)**
>
> In summary our contributions to address the reviewers concerns include,
>
> - Comparison of the proposed jailbreak method with other state-of-the-art (SOTA) methods in terms of Attack Success Rate (ASR) and Cost Per Prompt (CPP).
> - Enhancements to improve the presentation and readability of the paper.
> - Clarifications regarding the binary reward function used in our work and a through analysis of its comparison PAIR.
>
> We hope the technical clarifications, additional experiments, and justifications we provided have addressed your concerns. If you find the responses satisfactory, we kindly request you to consider raising the score above the acceptance threshold. Please let us know if there are any remaining issues or additional details you would like us to address, which we would do it right away.

---

> ### Comment · Reviewer_rSU9 · 2024-11-21
> **Response to Authors**
>
> **For Poor Readability**
>
> The author needs revise their manuscript, there are too many issues which the manuscript should not be published.
>
> **For Lack of Comparison**
>
> The CPP is a new concepts you mentioned during rebuttal. If you believe that the CPP can prove your method effectiveness, please revise in the pdf, instead of review response. I am still confused about what CPP is, as the authors only described it in an informal reply.
>
> **For Few experiments**
>
> The question is the same as above. The informal reply and the author's confusing expression make it difficult for me to understand what the authors did. Please modify it in the PDF. Thank you.
>
> **For Question 2**
>
> Please add citations to prove your point, which API uses filters, I don't think your statement is correct.
>
> **For Question 3**
>
> Thanks to the author for explaining this to me. Please add your clarification in the manuscript.
>
> ---
>
> **Before all my concerns addressed in revised pdf, I will not imporve my rating.**
>
> **In response, the author rearranged my review, making it very difficult for me to read.**

---

> ### Author Response · Authors · 2024-11-24
> **Manuscript Revision Details Part (1/2)**
>
> We sincerely thank the reviewer for their constructive feedback in improving our manuscript. Below, we address each of the concerns and highlight the corresponding updates made to the manuscript. For ease of reference, we provide the section and line numbers where the revisions have been implemented. All new revisions to the manuscript are highlighted in red font.
>
> ### **1. Poor Readability**
>
> **Reviewer Concern**: The introduction lacks clarity and relevance. Key concepts such as MAB, policy $\pi$, and action-value $Q$ are introduced abruptly without explanation. Algorithm 1 is not well explained.
>
> **Response**:
>
> - **Explanation of MAB**: We have added a detailed explanation of the multi-armed bandit (MAB) problem in the
>
>    Preliminaries Section (Section 3.1, Lines 175-184). This ensures readers are familiar with the concept before encountering its application in our work.
>
> - **Streamlined Introduction**: We have revised the manuscript to include a detailed section on preliminaries (Section 3.1),
>    where we thoroughly explain all the components necessary to understand our proposed method. Additionally, we have rewritten the introduction to improve its overall flow. This revision enhances clarity by clearly defining the problem we aim to solve and introducing the various components through toy examples, making the key concepts more intuitive and accessible to the reader.
>
> - **Clarification of Symbols**: Explanations for policy $\pi$ (Section 3.1, Lines 192-204), action-value $Q$ (Section 3.1, Lines 192-204),  and cost $C$ (Section 3.1, Lines 203-213) have been added in the Preliminaries section.
>
> - **Algorithm 1 Clarification**: We have clarified $E_{explore}$ and $E_{exploit}$ in the Preliminaries section
>    (Section 3.1, Lines 203–213), detailing their role in computing the cost function within our work. Additionally, we have included explicit explanations for these terms in Algorithm 1 (Lines 6 and 7). Furthermore, the SoC Attack Generation section has been rewritten to provide a clear, step-by-step explanation of both Algorithm 1 and Algorithm
>
> ### **2. Lack of Comparison**
>
> **Reviewer Concern**: Insufficient comparison with prior works and baseline jailbreak methods. CPP was introduced in the rebuttal but not in the manuscript.
>
> **Response**:
>
> - **Comparative Analysis**: We have included comparisons with state-of-the-art (SOTA) jailbreak methods in Section 5.5, evaluating our approach in terms of Attack Success Rate (ASR) and Cost Per Prompt (CPP). These metrics allow us to assess both the jailbreak performance and the time required to execute a jailbreak.
>
> - **CPP Definition**: CPP (Cost Per Prompt) is now defined and described in Section 5.5 (Lines 491-493). Its role in demonstrating the effectiveness of our method is explicitly stated.
>
> ### **3. Few Experiments**
>
> **Reviewer Concern**: Limited experimental analysis and lack of generalization tests.
>
> **Response**:
>
> - We have added additional experiments validating the generalization capability of our proposed method. We request the reviewer to see section 5.6 for the corresponding results.
> - We have also added experiments that compare the performance of the proposed judge function $J$ with SOTA judge functions from the Llama Guard Family in Section 5.1 Lines 411-412.
> - We further provide details about the hyperparameter setting of our proposed method in Appendix. C
> - We have also validated our method on LLMs of larger parameter size in Appendix. B
>
> ### **4. Question 2: Misstatement on LLM Safeguards**
>
> **Reviewer Concern**: The claim regarding classifiers in modern LLMs lacks citation.
>
> **Response**:
>
> - **Citations Added**: We have clarified our statement and added citations to justify the discussion of classifiers in LLM APIs    in Section 3.1 (Lines 161-163).
>
> ### **5. Question 3: Policy Optimization in Algorithm 1**
>
> **Reviewer Concern**: The policy update mechanism for $\pi$ and $\pi^\star$ is unclear.
>
> **Response**:
>
> - **Policy Update Explanation**: We introduce $\pi$ and $\pi^{*}$ and explain their connection to the   other key components of our proposed method in Section 3.1 (Lines 192–204). Furthermore, in Section 3.2, we elaborate on the relationship between $\pi$ and $\pi^{*}$. Finally, we have provided the mathematical equation representing the relation between $\pi$ and $\pi^{*}$ in Line 26 of Algorithm 1.

---

> > ### Author Response · Authors · 2024-11-24
> > **Manuscript Revision Details Part (2/2)**
> >
> > ### **6. Typos and Formatting Issues**
> >
> > **Reviewer Concern**: Typos and unclear references in the text and appendix.
> >
> > **Response**:
> >
> > - **Quotation Marks**: Quotation mark usage has been corrected in Section 3.1 (Lines 160–161).
> >
> > - **Binary Reward Statement**: The definition of the binary reward function has been restated for clarity in the Preliminaries section (Section 3.1, Lines 185–191).
> >
> > - **Appendix Formatting**: The term "Appendix" has been prefixed to relevant references to ensure clarity and consistency throughout the manuscript.
> >
> > We hope these revisions address your concerns. The updated manuscript reflects these changes, with all edits clearly marked for your review. We remain open to further suggestions and sincerely thank you for your valuable feedback. If these
> > updates satisfactorily resolve your concerns, we kindly request you to consider raising the score of our submission above the
> > acceptance threshold.

---

> > > ### Author Response · Authors · 2024-11-26
> > > **Reach Out to reviwer**
> > >
> > > Dear Reviewer rSU9,
> > >
> > > Thank you for your detailed feedback on our submission. We have carefully addressed all the concerns you raised in our rebuttal, providing clarifications, additional explanations, and updates to the manuscript where necessary.
> > >
> > > We kindly request you to engage with our rebuttal and let us know if there are any further clarifications, experiments, or improvements needed to address your concerns fully. Our aim is to ensure the manuscript meets the acceptance threshold, and we are committed to making any additional adjustments required to achieve this.
> > >
> > > Your engagement and input are invaluable to us, and we sincerely appreciate your time and effort in reviewing our work. Please let us know if there is anything more we can do to improve the submission.
> > >
> > > Thank you for your support.

---

> > > > ### Comment · Reviewer_rSU9 · 2024-11-26
> > > >
> > > > Dear Authors,
> > > >
> > > > Thanks for your clarification, I will increase the score.

---

> ### Author Response · Authors · 2024-11-27
> **Response To reviewer rSU9**
>
> Dear Reviewer rSU9,
>
> Thank you for your response and for increasing the score. We greatly appreciate your consideration.
>
> In addition to the experiments already included in the manuscript, we have conducted an additional experiment using Cohere, a closed-source LLM. For more details, you can refer to our reply to reviewer pzyz's reply titled **Response to Reviewer Suggestion** linked here : https://openreview.net/forum?id=jCDF7G3LpF&noteId=dLVoAFtkQV
>
>
> We would appreciate it if you could let us know if there are any further clarifications, experiments, or adjustments you would like to see in order to raise the score further, above the acceptance threshold. We are happy to make any additional improvements needed.
>
> Thank you once again for your valuable feedback and support.

---

> > ### Comment · Reviewer_rSU9 · 2024-11-27
> >
> > In my opinion, the revised version of the paper is marginally above the acceptance threshold.
> >
> > However, while the authors have made significant improvements to address the concerns of other reviewers, including the readability revisions I initially raised, the extent of the changes suggests that much of the paper has been reworked. I am uncertain whether such a substantial revision aligns with ICLR submission guidelines.
> >
> > **If the area chairs find this level of revision acceptable, I kindly ask that my score be considered as 6 when evaluating the paper for acceptance.**
> >
> > **The authors may wish to communicate this to the area chairs.**

---

### Official Review · Reviewer_uGpw · 2024-11-04

**Soundness:** 2
**Presentation:** 3
**Contribution:** 3
**Rating:** 8
**Confidence:** 3

**Summary:**

This paper proposes a novel jailbreaking attack paradigm named the sequence of contexts (SoC) attacks. By leveraging techniques in multi-armed bandit (MAB), this paper maximizes the likelihood of a successful jailbreaking attack (decided by CSQ). A theoretical upper bound for the gap between the obtained and optimal rewards is presented. Experimental results show that the proposed strategy indeed enhances the effectiveness of jailbreaking attacks.

**Strengths:**

**About novelty**

+ This paper studies jailbreaking attacks from an MAB perspective, bringing new insight into this area of research.

**About contribution**

+ This paper proposes a DMQ dataset that includes 3000 queries collected from previous works. The following works can use this as a benchmark.
+ The experimental results (figures 2 and 3) demonstrate that the proposed method enhances the effectiveness of jailbreaking methods, compared to the naive strategy.

**About presentation**

The presentation is clear. The algorithms and figures are well-made.

**Weaknesses:**

**About contribution**

+ This paper does not compare the proposed jailbreak attack with the existing methods.
+ The only theorem in this paper is almost trivial, and more importantly, the assumptions and limitations of this theorem are not discussed.
It is encouraging to include theoretical analysis in LLM research. However, the results are not strong enough to serve as "one of the main contributions" (as stated in Line 253) of an ICLR paper.

In brief, the authors claim three main contributions: creating a dataset, proposing a novel jailbreaking attack strategy, and providing a theoretical analysis. However, I think contributions (ii) and (iii) are slightly overstated. That is why I gave a score of 5 for this paper.

**About presentation**

+ This paper contains too many acronyms. I suggest adding a list of acronyms in the appendix. Besides, the authors do not provide a detailed explanation for some of the terms (e.g., sequence of context attack, and context switching queries) at their first appearance. It would make this paper more easy to follow if the authors add some explanations for the terms and reference them (e.g., see Section xxx for detailed discussions) when the term is mentioned for the first time.

+ The citation style (i.e., the green boxes) is dazzling.

**Questions:**

See the weakness part.

**Details Of Ethics Concerns:**

No ethics review is needed.

---

> ### Author Response · Authors · 2024-11-20
> **Rebuttal Part (1/5)**
>
> We sincerely thank you for your valuable comments, observations, and the time you have dedicated to evaluating our paper. We have carefully addressed your questions and the weaknesses you highlighted in a comprehensive manner. Our response is structured by presenting each weakness or question posed, followed by our detailed reply.
>
> **Weakness-1** This paper does not compare the proposed jailbreak attack with the existing methods.
>
> **Response-to-Weakness-1**
>
> In response to  reviewer GXrd, we detail and interpret the comparison with existing SOTA jailbreak methods. We request the reviewer to refer to the **Table RT 2** in **Weakness-3** and the corresponding response **Response-to-Weakness-3**. We additionally also detail and compare the existing jailbreak methods based on their computational costs.
>
> **Weakness-2**  The only theorem in this paper is almost trivial, and more importantly, the assumptions and limitations of this theorem are not discussed. It is encouraging to include theoretical analysis in LLM research. However, the results are not strong enough to serve as "one of the main contributions" (as stated in Line 253) of an ICLR paper.
>
> **Response-to-Weakness-2**
>
> Thank you very much for taking the time to review our work. Your feedback is invaluable in helping us refine and strengthen our research.
>
> We acknowledge that our theoretical results are derived within a standard Multi-Armed Bandit (MAB) framework, and some results, such as Lemma 1, leverage established techniques in MAB theory. We have cited the relevant literature in sections where these results are presented to provide context and clarity.
>
> We emphasize that the results in this paper are **original and tailored to our unique setting**, drawing inspiration from MAB literature but enriched with new insights to suit our framework.
>
> Unlike much of the existing ad hoc jailbreak literature in black-box settings, which lacks theoretical analysis, our work establishes a formal decision-making framework grounded in MAB theory. This principled approach provides predictive and explainable insights into the success of context-switching strategies, advancing beyond mere empirical observation.
>
> **To the best of our knowledge, this is the first application of MAB to a multi-step attack scenario using context-switching strategies in LLMs.** This novel approach introduces a structured decision-making framework for conducting multi-step SoC attacks, systematically exploring and exploiting context-switching queries (CSQs) to maximize jailbreak success in minimal steps, addressing a critical gap in the field.

---

> ### Author Response · Authors · 2024-11-20
> **Rebuttal Part(2/5)**
>
> ## Non-Triviality of Theorem 1 in the Context of SoC Attack Strategy
>
> In our Multi-Armed Bandit (MAB) framework with $\epsilon$-greedy exploration, deriving the bound $\mathbb{E}[N_i] \leq \frac{2 \log T}{\Delta_i^2}$ in the context of SoC attacks introduces unique challenges that differ from traditional applications. Specifically:
>
> - **Sequential Dependency in Rewards:** Although, we adopt a standard MAB setting, the SoC attack framework inherently introduces sequential dependencies in the rewards across CSQs. Each CSQ in the prompt sequence interacts with prior context switches, influencing the language model's responses. This sequential reward dependency deviates from the usual MAB assumption where each action’s reward distribution is independent of prior actions, making it non-trivial to derive tight bounds on the expected number of selections of suboptimal CSQs.
>
> - **Evolving Reward Gaps $\Delta_i$:** In SoC-based attacks, each selected CSQ subtly modifies the LLM's context, potentially altering its reward patterns over time. As a result, the gap $\Delta_i$ between the optimal and suboptimal CSQs may experience slight shifts throughout the sequence. This dynamic aspect introduces additional complexity in bounding the expected number of times suboptimal CSQs are selected, as it requires accommodating potential variations in $\Delta_i$ rather than assuming it remains constant, as in traditional MAB problems.
>
> Therefore, bounding $\mathbb{E}[N_i]$ in this context is non-trivial, as it extends the standard MAB theory by accounting for the cumulative effects of sequential context switching. This result is particularly important for optimizing SoC attack efficiency, ensuring minimal time is spent on ineffective CSQs.
>
> ### Limitations for Theorem-1
>
> #### Limitation 1: Rapid Convergence in Low Variance Settings
>
> Suppose all CSQs $i$ have reward variances close to zero, i.e., $\text{Var}(R_i) \approx 0$, with small reward gaps $\Delta_i$ between them. In such cases, the convergence to the optimal CSQ could be artificially accelerated.
>
> For a given $\Delta_i$, the bound $\mathbb{E}[N_i] \approx \frac{\log T}{\Delta_i^2}$ could become loose if $\Delta_i$ is very small, resulting in excessive early exploitation of suboptimal CSQs. This Limitation reveals that our bound depends heavily on the assumption of a sufficiently large reward gap $\Delta_i$, which may not hold in low-variance reward settings. This suggests a need for adaptive bounds that tighten based on observed variances.
>
> #### Limitation 2: Suboptimal CSQs with Near-Optimal Rewards
>
> Consider suboptimal CSQs $i$ for which $\Delta_i \to 0$ as $i$ approaches the optimal CSQ reward $\mu^*$. Here, the probability of selecting a suboptimal CSQ due to its close reward $\mu_i$ increases significantly. Mathematically, the probability of selecting a suboptimal CSQ is:
>
> $$\mathbb{P}(\hat{\mu}_i \geq \hat{\mu}^*) \leq 2 \exp\left(-\frac{n_i \Delta_i^2}{2}\right)$$
>
> As $\Delta_i \to 0$, $\exp\left(-\frac{n_i \Delta_i^2}{2}\right) \approx 1$, suggesting that suboptimal choices might persist even with large sample sizes. This reveals a limitation: for very small reward gaps, the theorem’s exponential convergence rate weakens, and the attacker may struggle to differentiate between near-optimal CSQs.
>
> #### Limitation 3: Misclassification in the Judge Function
>
> The judge function provides a binary reward $r_t \in \{0, 1\}$ based on whether a response is harmful. However, suppose the judge function misclassifies with an error rate $\delta$, giving the wrong binary classification with probability $\delta$. This can be modeled as an added noise term in the reward:
>
> $$\tilde{R}_t = R_t + \epsilon_t, \quad \epsilon_t \sim \text{Bernoulli}(\delta)$$
>
> The expected reward now has an error-adjusted bound:
>
> $$\mathbb{E} \left[ \sum_{t=1}^T \tilde{R}_t \right] \approx T \mu^* - O \left( \frac{K \log T}{\Delta_{\min}} + \delta T \right)$$
>
> where $\delta T$ represents the added regret due to judge misclassifications. This indicates that errors in the judge function directly affects the convergence and optimality of the CSQ selection process, requiring the attacker to account for these potential misclassifications when evaluating rewards.

---

> ### Author Response · Authors · 2024-11-20
> **Rebuttal Part(3/5)**
>
> ## Non-Triviality and Limitations of the Corollary in the SoC Attack Framework
>
> This corollary provides critical insight into the efficiency of the SoC attack generation process, specifically in terms of minimizing the number of selections needed to identify and consistently exploit the optimal CSQ for each Direct Malicious Query (DMQ). The Non-Triviality arguments and Limitations for this corollary are same as that of Theorem-1 as it is directly derived from it.
>
>  ## Non-Triviality and Limitations of the Lemma-2 in the SoC Attack Framework
>
> This lemma is non-trivial and significant for the following reason:
>
> - **Effect of Judge Misclassification on Cumulative Reward:** In SoC attack setting, the judge function can introduce misclassifications, leading to noisy rewards. Lemma 2 incorporates the judge misclassification rate $\delta$, accounting for the accumulated impact of such noise on the expected reward. This feature is crucial in our setting, as it formalizes how errors in reward signals impact the efficiency of the attack over multiple iterations. To the best of our knowledge, in LLM Jailbreaking setup, there does not exist any literature that studies the effect of misclassification of Judge function. This part makes it non trivial and an important contribution from a theoretical perspective
>
> ### Limitations for Lemma-2
>
> #### Limitation 1: High Judge Misclassification Rate ($\delta$ is Large)
>
> When the judge function’s misclassification rate $\delta$ is high, the correction term $\delta T$ increases, leading to a substantial deviation from the optimal reward:
>
> $$\mathbb{E} \left[ \sum_{t=1}^{T} \mu_{a_t} \right] \geq T \mu^* - O \left( \frac{K \log T}{\Delta_{\min}} + \delta T \right)$$
>
> In this scenario, the lemma highlights that a high misclassification rate by the judge function significantly lowers the expected total reward. This limitation underscores the importance of a highly accurate judge function to maintain effective SoC attacks, as increased noise in the reward signal can undermine the effectiveness of the algorithm.
>
> #### Limitation 2: Small Reward Gap ($\Delta_{\min} \to 0$)
>
> If the smallest reward gap $\Delta_{\min}$ approaches zero, the regret term $\frac{K \log T}{\Delta_{\min}}$ grows large:
>
> $$O \left( \frac{K \log T}{\Delta_{\min}} \right) \to \infty \quad \text{as} \quad \Delta_{\min} \to 0$$
>
> In this case, the lemma reveals that the algorithm may accumulate substantial regret, leading to a lower expected total reward. This limitation implies that the convergence of the total reward to the optimal reward is most effective when there is a significant reward gap between the optimal and suboptimal CSQs.
>
> #### Limitation 3: Early Stopping and Limited Episodes ($T$ is Small)
>
> For a small number of episodes $T$, the bound on the expected reward may not tightly approximate the optimal reward:
>
> $$\mathbb{E} \left[ \sum_{t=1}^{T} \mu_{a_t} \right] \geq T \mu^* - O \left( \frac{K \log T}{\Delta_{\min}} + \delta T \right)$$
>
> In this case, the bound shows that the algorithm requires a sufficient number of episodes to approach the optimal reward reliably. For short episodes or early stopping, the cumulative effect of suboptimal choices and judge misclassifications may result in a lower reward, affecting the efficiency of SoC attacks in limited settings.

---

> ### Author Response · Authors · 2024-11-20
> **Rebuttal Part (4/5)**
>
> **Weakness-3:** In brief, the authors claim three main contributions: creating a dataset, proposing a novel jailbreaking attack strategy, and providing a theoretical analysis. However, I think contributions (ii) and (iii) are slightly overstated. That is why I gave a score of 5 for this paper.
>
>  **Response-to-Weakness-3:**
>
> Thank you for your valuable feedback. We appreciate the opportunity to clarify the significance of our contributions, particularly contributions (ii) and (iii), which focus on the novel attack strategy and its theoretical underpinnings.
>
> While we understand that many existing black-box jailbreaking methods focus primarily on the practical success of prompting techniques, we believe our approach provides a structured, explainable framework that addresses a critical gap in current research. Here are key points that highlight the unique advantages and value of our MAB-based SoC   attack methodology:
>
>
> 1. **Explainability and Insight into Attack Dynamics:** Our work frames the problem within a MAB setup, enabling a theoretically grounded analysis of context-switching decisions. Unlike existing methods, that rely on trial-and-error prompt engineering, our theoretical results explain why certain sequences succeed in jailbreaking, providing insights that are valuable for both refining attacks and designing effective defenses.
>
> 2. **Efficiency and Theoretical Rigor in Strategy Optimization:** By leveraging the $\epsilon$-greedy MAB framework, our approach quickly converges to optimal context-switching strategies with minimal exploratory overhead. The derived bounds, such as limiting suboptimal CSQ selections, enhance the attack's efficiency and effectiveness. These theoretical contributions provide a rigorous foundation to explain the empirical success of our method, addressing a gap in the largely ad-hoc approaches dominating current jailbreaking literature. See Response, **Response-2** to reviewer **UGpw** for more details.
>
>
> **Weakness-4:** This paper contains too many acronyms. I suggest adding a list of acronyms in the appendix. Besides, the authors do not provide a detailed explanation for some of the terms (e.g., sequence of context attack, and context switching queries) at their first appearance. It would make this paper more easy to follow if the authors add some explanations for the terms and reference them (e.g., see Section xxx for detailed discussions) when the term is mentioned for the first time.
>
>
>
> **Response-to-Weakness-4)**
>
> We thank the reviewer for their constructive suggestion to improve the readability and presentation of the paper! We agree that, this will improve the paper. We will add the following **Table RT4** in the appendix section and will modify the text of the main paper to incorporate the reviewer’s suggestion for providing references to tie the various sections of the paper together.
>
>
>
> | **Acronym** | **Definition**                      |
> |-------------|-------------------------------------|
> | LLM         | Large Language Models               |
> | SoC         | Sequence of Contexts Attack         |
> | CSQ         | Context Switching Query             |
> | DMQ         | Direct Malicious Query              |
> | MAB         | Multi Arm Bandit                    |
> | ASR         | Attack Success Rate                 |
> | CPP         | Cost Per Prompt                     |
> | UAT         | Universal Adversarial Triggers      |
> | App.        | Appendix                            |
> | Fig.        | Figure                              |
> | Algo.       | Algorithm                           |
>
> **Table RT4 : Table of Acronyms**
>
> **Weakness-5** The citation style (i.e., the green boxes) is dazzling.
>
> **Response-to-Weakness-5)**  Thanks for the suggestion! We will modify the color of the citations in the main paper and appendix.

---

> ### Author Response · Authors · 2024-11-20
> **Rebuttal Part(5/5)**
>
> In response to the reviewer’s concerns, our contributions include:
>
> - A comparison of the proposed jailbreak method with SOTA methods in terms of ASR and CPP.
> - Clarification of the significance of Theorem 1, emphasizing the complexities involved in its derivation, and a discussion of its potential limitations to provide a comprehensive understanding of our theoretical contributions.
> - Presentation of Lemma 2 as a critical result in SoC attacks, offering a bound on expected cumulative reward while accounting for exploration costs and judge misclassification errors. This analysis highlights practical limitations such as high misclassification rates, small reward gaps, and limited episode lengths, essential for understanding the efficiency of SoC attacks.
> - Enhancements to improve the presentation and readability of the paper.
>
> We hope the technical clarifications, additional experiments, and justifications we provided have addressed your concerns. If you find the responses satisfactory, we kindly request you to consider raising the score above the acceptance threshold. Please let us know if there are any remaining issues or additional details you would like us to address, which we would do it right away.

---

> > ### Author Response · Authors · 2024-11-26
> > **Reach Out to reviewer uGpw**
> >
> > Dear Reviewer uGpw,
> >
> > Thank you for your detailed feedback on our submission. We have carefully addressed all the concerns you raised in our rebuttal, providing clarifications, additional explanations, and updates to the manuscript where necessary.
> >
> > We kindly request you to engage with our rebuttal and let us know if there are any further clarifications, experiments, or improvements needed to address your concerns fully. Our aim is to ensure the manuscript meets the acceptance threshold, and we are committed to making any additional adjustments required to achieve this.
> >
> > Your engagement and input are invaluable to us, and we sincerely appreciate your time and effort in reviewing our work. Please let us know if there is anything more we can do to improve the submission.
> >
> > Thank you for your support.

---

### Official Review · Reviewer_GXrd · 2024-11-04

**Soundness:** 3
**Presentation:** 2
**Contribution:** 3
**Rating:** 6
**Confidence:** 3

**Summary:**

This paper proposes a novel Sequence of Context (SoC) jailbreak attack that leverages Multi-Armed Bandit (MAB) to automatically guide context selection. The authors provide an in-depth theoretical analysis of the upper bound on the expected sequence length. Experimental results demonstrate the effectiveness of the proposed method in jailbreaking language models.

**Strengths:**

1. The use of MAB for automated context selection in jailbreaking is novel
2. The theoretical derivation of the upper bound for the SoC attack length is well-established
3. The experimental results effectively demonstrate the method's efficacy

**Weaknesses:**

1. Compared to other automatic jailbreak attacks (e.g., GCG, PAIR), this method requires dataset collection and policy model training, making it more resource-intensive and time-consuming
2. The proposed method is limited to pre-defined harmful query categories, and its extensibility to unseen categories is not thoroughly investigated
3. The paper lacks comparison with state-of-the-art jailbreak attacks in terms of attack success rate and computational cost

**Questions:**

1. Could you explain the necessity of including the direct malicious query (DMQ) in the context? Given that language models with safety alignment typically reject DMQs, would it be possible to remove DMQ from the context to reduce context length?
2. How does the proposed judgment method compare with widely-used judgment systems (e.g., Llama Guard family) that are more common in jailbreak literature?

---

> ### Author Response · Authors · 2024-11-20
> **Rebuttal Part (1/3)**
>
> We sincerely thank you for your valuable comments, observations, and the time you have dedicated to evaluating our paper. We have carefully addressed your questions and the weaknesses you highlighted in a comprehensive manner. Our response is structured by presenting each weakness or question posed, followed by our detailed reply.
>
>
> **Weakness-1 :** Compared to other automatic jailbreak attacks (e.g., GCG [4], PAIR [2]), this method requires dataset collection and policy model training, making it more resource-intensive and time-consuming
>
> **Response-to-Weakness-1 :**
> -	The proposed method reformulates the multi-turn jailbreak problem as a sequence of context-switching actions aimed at steering an LLM into a jailbroken state with minimal steps using a multi-arm-bandit framework.
> -	A policy is learned to determine the optimal action at each step by evaluating the action values based on binary rewards from previous steps. The policy uses an ε-greedy strategy to balance exploiting known actions with maximum value and exploring new actions for potentially better outcomes.
> -	Our proposed method jailbreaks the LLM in lesser time than state-of-the-art (SOTA) methods due to the policy training stage, as evidenced by average time taken for jailbreak (cost-per-prompt, CPP) reported in Table RT2.
> - To enable fair comparisons across LLM models, a dataset of context-switching prompts is created for evaluation. Alternatively, context-switching prompts can be dynamically generated at each step, at the cost of an additional LLM API call per step.
> - Unlike white-box methods, which rely on accessing LLM parameters and gradients to search for adversarial inputs prompts, the proposed method requires only API calls, making it computationally less expensive. This results in improved performance w.r.t to CPP, as demonstrated in Table RT2.
> - The method uses Multi-Armed Bandits (MAB) to efficiently learn optimal context-switching sequences, requiring an initial training phase but enabling generalization across malicious query categories. Unlike GCG or PAIR, it is mathematically robust (Theorem 1), minimizes unnecessary steps, and ensures scalable, efficient jailbreaks with significant computational benefits.
>
> **Weakeness-2 :** The proposed method is limited to pre-defined harmful query categories, and its extensibility to unseen categories is not thoroughly investigated
>
> To evaluate the generalizability of our approach, we perform 5-fold cross-validation using 18 harmful query categories (DMQs) which are mentioned in the main paper in section 3.1. In each cross validation fold, the proposed method is trained on a subset of these DMQ (harmful-query-categories) categories and tested on unseen ones. The Attack Success Rate (ASR) for each fold are reported in Table RT1.
>
> | **Train DMQ Categories** | **Test DMQ Categories** | **ASR** |
> |-----------------------------------------------------------------------------------------------------------------------------------------------------------------------------------------------------------------------------------------------------------------------------------------------------------------------|----------------------------------------------------------------------------|----------|
> | 1, 2, 3, 4, 5, 6, 7, 9, 10, 11, 15, 16 | 8, 13, 14, 17, 18 | 0.905 |
> | 2, 3, 5, 6, 9, 10, 11, 12, 14, 15, 16, 17 | 1, 4, 7, 8, 13, 18 | 0.933 |
> | 1, 2, 3, 5, 6, 7, 9, 10, 11, 12, 14, 15 | 4, 8, 16, 18, 13, 17 | 0.921 |
> | 1, 2, 3, 5, 6, 7, 10, 11, 12, 13, 15, 16 | 1, 2, 9, 11, 14, 18 | 0.925 |
> | 1, 2, 3, 5, 6, 7, 8, 10, 12, 14, 15, 18 | 4, 13, 16, 9, 17, 11 | 0.923 |
> | **Average Cross-Validation ASR** | | **0.9214** |
> **Table RT1:** Generalization metrics for unseen DMQ categories using 5-fold cross-validation across four LLM models.
>
> The cross-validation results demonstrate the robustness and generalizability of the approach, with the Attack Success Rate (ASR) consistently high (0.905 to 0.933, average 0.9214), showing effective generalization to unseen harmful query categories.

---

> > ### Author Response · Authors · 2024-11-20
> > **Rebuttal Part (2/3)**
> >
> > **Weakness-3 :**  The paper lacks comparison with state-of-the-art jailbreak attacks in terms of attack success rate and computational cost
> >
> > **Response-to-Weakness-3:**
> >
> > | **Method** | **ASR (%)** | CPP(s)
> > |------------------------|-------------|-------------------|
> > | ReNeLLM [1] | 0.47 |132s
> > | AUTODAN [3] | 0.38| 955s
> > | PAIR [2] | 0.35| 146s
> > | GCG [4] | 0.32| 564s
> > | Proposed SoC Attack | 0.95 | 15s
> > **Table RT2:** Comparison of Attack Success Rates (ASR) and Cost Per Prompt(s) for Various Jailbreak Methods.
> >
> > The table highlights the superior performance of the proposed SoC attack compared to existing jailbreak methods. It achieves the highest Attack Success Rate (ASR) of 95%, significantly outperforming methods like ReNeLLM (47%), AUTODAN (38%), PAIR (35%), and GCG (32%). Additionally, the SoC attack has the lowest cost per prompt (CPP) at just 15 seconds, demonstrating both exceptional efficiency and effectiveness over other state-of-the-art approaches.
> >
> > The GCG [4] method performs attacks by optimizing adversarial suffixes/prefixes to manipulate the LLM's output probability distribution, requiring gradient computations over LLM parameters and the use of computationally expensive coordinate descent. In contrast, the proposed approach relies solely on API access, making it significantly more efficient. Templatized methods like ReNeLLM [1] and PAIR [2] depend on manual calibration and iterative prompt refinement via auxiliary LLMs, lacking a decision-making framework and theoretical guarantees, unlike the proposed method.
> >
> > **Question-1:**  Could you explain the necessity of including the direct malicious query (DMQ) in the context? Given that language models with safety alignment typically reject DMQs, would it be possible to remove DMQ from the context to reduce context length?
> >
> > **Response-to-Question-1:**
> >
> > Direct malicious queries (DMQs) are simple prompts designed to elicit harmful responses from LLMs. As the reviewer correctly pointed out, due to safety alignment, LLMs typically resist responding to such straightforward inputs. DMQs are usually short (10-20 tokens), so their removal from the context minimally affects the overall context length. Context-switching queries (CSQs), on the other hand, build context around the DMQ, providing additional information to steer the LLM toward a jailbroken state. These queries are crafted to bypass safety filters and ethical alignment without revealing clear harmful intent. Thus, removing the DMQ from the context will lead to the generation of unaligned responses that fail to coherently address the DMQ due to the lack of a harmful intent in the CSQs.
> >
> > **Question-2:**  How does the proposed judgment method compare with widely-used judgment systems (e.g., Llama Guard family) that are more common in jailbreak literature?
> >
> > **Response-to-Question-2:**
> >
> > The evaluation compares the Llama-Guard models (Llama-3-Guard-1B and Llama-3-Guard-8B) with the judge function from the main paper using the same dataset [5], containing human-annotated labels for toxic and non-toxic classes. Binary classification metrics—Precision, Recall, F1-Score, and Accuracy—are used to compare predicted labels against ground truth. The results, summarized in Table RT3, highlight the performance differences among these approaches.
> >
> > | **Metric**   | **Llama Guard 1B** | **Llama Guard 8B** | **Our Judge** |
> > |--------------|---------------------|---------------------|------------------|
> > | Precision    | 0.68              | 0.96              | 0.95           |
> > | Recall       | 0.99              | 1.00              | 1.00            |
> > | F1 Score     | 0.81              | 0.98              | 0.97           |
> > | Accuracy     | 0.76              | 0.97              | 0.97            |
> > **Table RT3:** Comparison of the proposed judge function and the judge function in the Llama-Guard-Family.
> >
> >
> > The proposed judge function matches the **Llama Guard 8B** in its Recall (1.00) and Accuracy (0.97) and achieves comparable Precision (0.95) and F1 Score (0.97). Compared to **Llama Guard 1B**, **Our Judge** demonstrates a significant advantage, outperforming it in Precision (0.95 vs. 0.68), F1 Score (0.97 vs. 0.81), and Accuracy (0.97 vs. 0.76). This highlights the robust and consistent performance of the proposed judge function. Additionally, using the proposed judge function allows us to define custom categories using the input prompt template. We have provided more details about this in the section B.4 in the appendix section of the main paper.
> >
> > **References :**
> > - [1] ReNeLLM : https://arxiv.org/pdf/2311.08268
> > - [2] PAIR : https://arxiv.org/pdf/2310.08419
> > - [3] AUTODAN :  https://arxiv.org/pdf/2310.04451
> > - [4] GCG : https://arxiv.org/pdf/2307.15043
> > - [5] ToxiGen : https://arxiv.org/abs/2203.09509

---

> > > ### Author Response · Authors · 2024-11-20
> > > **Rebuttal Part (3/3)**
> > >
> > > In summary our additional contributions to address the reviewer's concerns include
> > >
> > > - Comparison of the proposed jailbreak method with other state-of-the-art (SOTA) methods in terms of Attack Success Rate (ASR) and Cost Per Prompt (CPP).
> > > - Analysis and validation of the proposed judge function in comparison to SOTA judge functions within the LLama-Guard family.
> > > - Examination of the generalization capabilities of the proposed jailbreak attack when applied to unseen harmful query categories.
> > > - Clarification and requirements regarding the inclusion of Direct Malicious Queries (DMQ) in the context provided to the LLM.
> > >
> > > We hope the technical clarifications, additional experiments, and justifications we provided have addressed your concerns. If you find the responses satisfactory, we kindly request you to consider raising the score above the acceptance threshold. Please let us know if there are any remaining issues or additional details you would like us to address, which we would do it right away.

---

> > > > ### Author Response · Authors · 2024-11-26
> > > > **Reach Out to Reviewer GXrd**
> > > >
> > > > Dear Reviewer GXrd,
> > > >
> > > > Thank you for your detailed feedback on our submission. We have carefully addressed all the concerns you raised in our rebuttal, providing clarifications, additional explanations, and updates to the manuscript where necessary.
> > > >
> > > > We kindly request you to engage with our rebuttal and let us know if there are any further clarifications, experiments, or improvements needed to address your concerns fully. Our aim is to ensure the manuscript meets the acceptance threshold, and we are committed to making any additional adjustments required to achieve this.
> > > >
> > > > Your engagement and input are invaluable to us, and we sincerely appreciate your time and effort in reviewing our work. Please let us know if there is anything more we can do to improve the submission.
> > > >
> > > > Thank you for your support.

---

> > > > > ### Comment · Reviewer_GXrd · 2024-11-27
> > > > > **Thank you for your detailed responses!**
> > > > >
> > > > > Thank you for the detailed responses! All of my concerns are addressed, and I will increase my score. Good luck on your submission!

---

### Author Response · Authors · 2024-11-24
**Response to all Reviewers**

**Part 2/2**


- **Clarification on API Filters for Direct Malicious Queries (DMQ):**  In response to the reviewer's comment, we have added relevant citations in **Section 3.1 (Preliminaries)** to support our statement regarding the use of filters in APIs to direct malicious queries (DMQ). These citations clarify the role of filters in mitigating harmful or malicious inputs in API systems.  We believe this additional information strengthens our argument, and we invite the reviewer to refer to the updated section for a more comprehensive explanation.

- **Testing on Larger LLMs:**
  We appreciate the reviewer's feedback regarding the evaluation of our work on only Mistral and Llama models, particularly with the focus on smaller LLMs (up to 8B). In response, we have expanded our experiments to include larger models, specifically Llama-13B and Vicuna-13B, to better demonstrate the performance of our approach. The results for these models can be found in **Appendix B**.  We believe these additional experiments strengthen the validity of our findings and invite the reviewer to refer to the updated results for a more comprehensive evaluation.


All new changes have been highlighted in **red-font**. We believe these revisions strengthen the manuscript and address all concerns. Please let us know if further adjustments are needed.

Given these improvements, we kindly request that the score be reevaluated with consideration for raising it above the acceptance threshold. We sincerely appreciate your time and effort in reviewing our work and look forward to your feedback on the revised draft.

We invite all reviewers to engage with our rebuttal and modifications to the draft.

---

### Author Response · Authors · 2024-11-24
**Response to all Reviewers**

**Part 1/2**

We would like to sincerely thank all the reviewers for their valuable comments and suggestions. We have carefully considered all feedback and have incorporated the recommended changes into the revised draft.


- **Comparison to State-of-the-Art (SOTA) Jailbreak Attacks:**
In response to the reviewers' request, we have conducted experiments comparing our proposed method to SOTA automatic jailbreak attacks, evaluating both the Attack Success Rate (ASR) and Cost Per Prompt.
  - We kindly direct the reviewers to **Section 5.5** for the relevant updates.
  - A detailed explanation of the **ASR metric** is available in **Section 5.2**, while the **cost-per-prompt metric** is described in **Section 5.5**.

- **Analysis of the Generalization Capability:**
In line with the reviewer's suggestion, we have conducted an analysis of the generalization capability of our proposed method to unseen DMQ categories. The results are presented in **Section 5.6** of the revised draft.

- **Comparison of the Proposed Judgment System to State-of-the-Art (SOTA) Judgment Systems:**
We compare our proposed judgment system with SOTA judgment systems from the Llama-Guard family in **Appendix D.5**.

- **Analysis of the Limitations and Non-Triviality of Our Theoretical Contributions:**
We have analyzed various scenarios where our proposed theoretical results are not tight and have illustrated these cases in **Appendix H**. Furthermore, the broader significance of our contributions to the LLM-Safety community is also discussed in **Appendix H**.

- **Acronyms:**
  For convenience, a table listing all acronyms used in this paper can be found in **Appendix A**.

- **Citation Style:**
  To improve readability, we have updated the citation color in the revised draft.

- **Clarification of Multi-Armed Bandit (MAB) and Preliminaries:**
  In response to the reviewer's comment, we have revised the introduction to provide a clear explanation of Multi-Armed Bandit (MAB), ensuring that the concept is properly introduced before being referenced in the abstract and **Figure 1**. We request the reviewer to refer to the revised introduction section. Additionally, we have included a detailed explanation of the MAB method in **Section 3.1**, along with clarifications of the associated notation. Furthermore, we have also illustrated all the necessary preliminaries throughout the manuscript to ensure that readers can easily follow the content without confusion.

- **Improvement of the Introduction Structure:**
  We have restructured the introduction to improve readability and relevance. We removed unnecessary content and streamlined the explanation to focus on the key contributions of the paper. Specifically, we have clarified the third paragraph and provided more details to avoid introducing too many concepts without adequate context.


- **Response to White-Box Attack Citations:**
  We appreciate the reviewer's suggestion regarding the inclusion of additional white-box attack references. In response, we have added new citations in **Section 2 (Related Work)**, covering a broader range of white-box attacks, such as fine-tuning attacks, improved GCG, and interpretability-based approaches. We believe that our citations are relevant and closely tied to the context of our work. However, if the reviewer finds any specific reference to be problematic or irrelevant, we would be happy to reconsider and make the necessary adjustments. We invite the reviewer to highlight any particular citation that may not align with the focus of the paper.

---

### Author Response · Authors · 2024-11-26
**Reach out to All the Reviewers**

Dear Reviewers,

Thank you for taking the time to review our submission and provide your valuable feedback. We have carefully addressed all the comments and concerns raised during the review process, as detailed in our rebuttal.

We kindly request you to review our responses and let us know if there are any remaining clarifications or additional experiments that you believe are necessary to improve the manuscript. Our goal is to ensure that our work meets and exceeds the acceptance threshold, and we are committed to making any further improvements required to achieve this.

Your input is invaluable in helping us refine our work, and we deeply appreciate your engagement and support. Please feel free to let us know if there is anything else we can address to strengthen the submission.

---

### Author Response · Authors · 2024-12-03
**Message of Appreciation**

Dear Reviewers, ACs and PCs,

Thank you for your valuable feedback and insights which helped improve our work. We deeply appreciate the time, effort, and expertise you dedicated to the peer review process.

Best regards,
Authors #13287

---

### Meta-Review · Area_Chair_Yx65 · 2024-12-13

**Metareview:**

The paper proposes a new Sequence of Context (SoC) attacks for LLMs, and It is novel and interesting to transform the original problem to multi-armed bandit (MAB) optimization problem. After the author's rebuttal, Almost all the reviewers increase the score, and agree to accept it. But the theory appeared in this paper is not new. So I recommend this paper as a poster.

**Additional Comments On Reviewer Discussion:**

After the author's rebuttal, Almost all the reviewers increase the score

---

### Decision · Program_Chairs · 2025-01-22

Accept (Poster)